# National Dog Survey: Describing UK Dog and Ownership Demographics

**DOI:** 10.3390/ani13061072

**Published:** 2023-03-16

**Authors:** Katharine L. Anderson, Rachel A. Casey, Ben Cooper, Melissa M. Upjohn, Robert M. Christley

**Affiliations:** Dogs Trust, 17 Wakley Street, London EC1V 7RQ, UK

**Keywords:** dog, ownership, demographics, United Kingdom

## Abstract

**Simple Summary:**

Dog ownership is common in the United Kingdom; however, current dog ownership demography data are lacking. Understanding our pet dog population is paramount to meeting the health and welfare needs of the population. Therefore, this paper provides an overview of a National Dog Survey, aiming to better understand and provide a snapshot of the current dog population in the UK. The results of this paper provide up-to-date demographic data for both dogs and their owners and highlight patterns and trends both long-term and more recently, since the COVID-19 pandemic. Monitoring of trends and patterns of the dog population and dog owners going forward is needed in order to continually identify the needs and welfare state of the population, enabling those working in the canine welfare field to provide appropriate support and services.

**Abstract:**

With dogs being the most commonly owned companion animal in the United Kingdom, knowledge about dog demographics is important in understanding the impact of dogs on society. Furthermore, understanding the demography of dog owners is also important to better target support to dogs and their owners to achieve optimal welfare in the canine population. Combining natural fluctuations in the population and unprecedented events such as the COVID-19 pandemic, the need for an up-to-date large-scale dataset is even more paramount. In order to address this, Dogs Trust launched the ‘National Dog Survey’ to provide a large population-level dataset that will help identify key areas of concern and needs of owners and their dogs. The online survey was completed by a total of 354,046 respondents owning dogs in the UK, providing data for 440,423 dogs. The results of this study highlight dog demographics, including acquisition and veterinary factors, as well as owner demographic and household information. Finally, general trends in ownership, and more specifically those following the COVID-19 pandemic, are described. This paper’s findings provide valuable insight into the current population of dogs and their owners in the UK, allowing for the most appropriate products, services, interventions and regulations to be developed, reducing the likelihood of negative welfare outcomes such as health and behaviour issues, relinquishment or euthanasia. Furthermore, with significant changes to the dog population following the COVID-19 pandemic highlighted, this dataset serves as an up-to-date baseline for future study comparisons to continue to monitor trends and patterns of the dog population and dog owners going forwards.

## 1. Introduction

Providing companionship, love and affection to owner(s) are reported as common reasons for pet ownership in the United Kingdom (UK), with pets often considered members of the family [1,2]. Knowledge about pet and owner demographics is therefore important in understanding the impact of pets on society. Dogs are the most commonly kept companion animal in the UK, with an estimated 34% of households owning one or more dogs [3]. Whilst attempts to estimate the UK dog population (including the unowned population) exist [3,4,5,6] they are not without limitations such as sampling biases, sample size, geographical representation and generalisability. The current population estimate following the COVID-19 pandemic in the UK is around 13 million dogs [3]. However, there are limited data on the current demography of the dog-owning population within the UK [4,5,7,8] or the wellbeing of the dog population [9]. Current and accurate dog ownership population data are needed to maintain and improve the health and welfare of the population, from a public and One Health perspective. As dogs are dependent on their owners to live a rich and fulfilled life, understanding the demography of dog owners is also important to better target interventions, products and support to dogs and their owners to achieve optimal welfare of the canine population.

Due to the broad application of these data, understanding the canine population has implications for many stakeholders working in the field. For example, for those within the veterinary industry, understanding the demographics of dog ownership is important for provision of and access to health and behaviour services that adequately meet the demands and needs of the population. Having accurate and up-to-date data also aids those involved in developing and predicting the outcomes of interventions and services such as welfare organisations, as well as in the development of protective regulations and legislations surrounding dogs and their owners. These data are also valuable to researchers in the surveillance of the health and welfare of the population, whilst guiding future studies and hypotheses and acting as a baseline for future comparisons to monitor trends.

Given their importance to a wealth of stakeholders, there has been long-term interest in obtaining demographic and population data regarding pet ownership within many countries worldwide including the USA [10,11,12,13], Brazil [14], Canada [15], Germany [16], Italy [17,18], Mexico [19], New Zealand [20], Australia [21], Portugal [22], Ireland [23] and the UK [4,5,7,8,24,25,26]. Each year, the People’s Dispensary for Sick Animals (PDSA) release their Pet Animal Wellbeing (PAW) report, summarising data regarding the needs and key welfare issues facing the current UK pet population [9]. Similarly, the Pet Food Manufacturers Association (PFMA) runs an annual survey reporting on data regarding the UK pet population [3]. As existing UK data focus on pet ownership in general, and therefore include a broad number of species, sample sizes related specifically to dogs and dog ownership are limited, with samples between 250 and 5000 dogs/owners. In order to delve deeper into the dog population and their owners there is a need for a larger canine-only dataset. Furthermore, with changes to pet populations and patterns of ownership over time [8,27], it is important to have up-to-date data for this population. This is increasingly important in light of COVID-19 lockdown restrictions, from March 2020 in the UK, resulting in huge changes for the UK pet dog population and a reported boom in dog ownership [28]. Combining natural fluctuations of the population and unprecedented events such as the COVID-19 pandemic, the need for an up-to-date large-scale dataset is even more paramount.

As the UK’s largest canine welfare charity, Dogs Trust aims to ‘make the world a safe and happy place for dogs’ [29] and its activities aim to help the dog-owning population to ensure optimal canine welfare. Dogs Trust launched the ‘National Dog Survey’ to provide a large population-level dataset that will help identify key areas of concern and needs of owners and their dogs. Here we describe the methodology of conducting a large-scale national survey and report the demographic data from the survey. As well as describing the current characteristics of the UK dog and dog-owning population, these data can aid hypothesis generation and act as baseline data for future research on, and intervention development targeted at, the dog population and their owners. Specifically, this paper aims to (1) identify the characteristics of dog-owning households from a large cross-sectional study, describing the dog and owner demographics of the UK dog population as of October 2021, and (2) describe changes in the demographics of ownership and dogs over time and since the COVID-19 pandemic.

## 2. Materials and Methods

Ethical approval for this study was granted by the Dogs Trust Ethical Review Board, reference number ERB050.

### 2.1. Questionnaire Development

The data collection tool for this study was an online survey developed by Dogs Trust teams (Marketing, Communications, Digital and Research) with the questions guided through consultation with the Dogs Trust leadership team based on the charity’s key strategic areas of interest. The survey was developed alongside a creative agency (https://www.goodagency.co.uk/ [accessed on 15 March 2023]) and hosted on its own domain (nationaldogsurvey.org.uk) for the duration of the study period (developed and built by a digital experience agency (https://manifesto.co.uk/ [accessed on 15 March 2023]). Regular Dogs Trust branding was used throughout the survey and during advertising. Question areas included within the survey were as follows:IntroductionAbout your dogYour dog’s behaviourYour dog plans and thoughtsAdditional questionsAbout you

Question formats included a combination of multiple choice, demographic, open-ended and Likert-scale attitude-based questions.

### 2.2. Study Participants and Data Collection

The target population for this study was dog-owning adults (≥18 years) in the UK, including both new dog owners (during the pandemic) and existing dog owners. Households with more than one dog were asked to submit information initially for the most recently acquired dog (although information about other dogs could be added later), and only one survey per household was requested.

A convenience volunteer sampling technique was used to collect responses from participants willing to volunteer their time to take part in the study. Recruitment took place via a very broadly distributed awareness campaign. At the outset, participants were invited and recruited to take part via the sharing of marketing material with Dogs Trust supporters through outlets they currently interacted with, such as via Dogs Trust social media accounts and webpages. Incremental activity was then used throughout the period the survey was live to target a far wider population, including hard-to-reach audiences not already engaging with Dogs Trust. Incremental recruitment activity aiming to maximise awareness of the study included multiple methods of promotion and advertising using national TV, online, radio and magazine advertisements and social targeting of advertisements (targeted towards specific demographic groups). Members of the public attending certain dog events, as well as city-centre activation events, were also invited to participate in the study during the events (by completing the survey on an iPad). As this was a ‘census’ style descriptive project, seeking maximum possible coverage and responses, it would be inappropriate to use typical techniques used to determine the sample size, and therefore a projected aim of 200,000 responses was anticipated.

The survey was conducted online using software hosted on the web domain ‘nationaldogsurvey.org.uk’ (accessible on both desktop and mobile devices during the period the survey was live), taking about 10 min to complete. The survey ran from 10 September 2021 to 25 October 2021.

### 2.3. Data Analysis

Once the survey had closed, data were downloaded from the server and imported into R (version 4.1.3; [30]) for cleaning. Data were provided by Manifesto in three CSV files, comprising answers relating to owners, non-owners and dogs, respectively. Duplicate entries with the same ID code (160 responses) were identified and removed.

As data cleaning proceeded, data were organized into smaller tables based on analysis concepts (such as human/dog demographics, behaviour, expectations, etc.). Data cleaning included, but was not limited to, the following steps:Free text entries in “number of dogs in household” changed to numeric, and numbers greater than 15 removed as implausiblePostcode trimmed to first section where the full postcode was providedMultiple choice responses shortened and spaces replaced with underscoresQuestions where respondents gave “Less”, “Same” or “More” as a response re-coded to −1, 0 and 1 to aid plotting and analysisNA responses for a given question removed, whilst retaining complete answers for the same userRemoval of respondents who withdrew consent for data use

Following cleaning, descriptive statistics and summaries (counts and percentage) for all dog and owner demographic variables were collated. Variables of interest were compared with cross tabulation, and associations were tested using Chi-square and one-sample proportion tests where appropriate.

## 3. Results

A total of 354,046 respondents owning dogs in the UK completed the survey and their full responses were available for the analysis in this study. Respondents were asked to complete the survey first for their most recently acquired dog and were later given the opportunity to answer for any other dog(s) they owned, providing data for 440,423 owned dogs in total. With the population estimated to be 13 million dogs, this represents around 3.4% of the canine population in the UK in this sample. Demographic summaries for dogs (Appendix A) and owners (Appendix A) were compiled.

### 3.1. Dog Demographics (Appendix A)

The majority of dogs in this sample were young adults aged between 3 and 6 years. The mean age of dogs was 5 years and 11 months and the median 5 years and 5 months. There were greater numbers of males (54%) than females (46%); a one-sample proportion test indicated that these proportions significantly differed from 0.5 (X-squared = 2843.6, df = 1, *p*-value < 0.001).

Regarding neutering, 72% of dogs were neutered, and neutering was significantly more common in females than males overall (76% and 69%, respectively; Pearson’s X-square = 2872.9, df = 1, *p*-value < 0.001, odds ratio (OR) 1.37 (95% CI 1.35–1.39)). When considering age, females continued to be significantly more likely to be neutered than males across all age groups. In young dogs under the age of 2, females had an odds ratio of 1.16 compared to males (Appendix A; X-squared = 79.641, df = 1, *p*-value < 0.01, OR 1.16 (95% CI 1.12–1.19), and in older dogs greater than 10 years females had 2.48 times the odds of being neutered compared to males (Appendix A; X-squared = 1538.3, df = 1, *p*-value < 0.01, OR 2.48 (95% CI 2.36–2.59) respectively). For both females (X-squared = 37956, df = 19, *p*-value < 0.01) and males (X-squared = 32450, df = 19, *p*-value < 0.01) the likelihood of neutering differed significantly with age (Appendix A). Figure 1 shows a population structure based on sex and neuter status. Dogs were increasingly neutered around 1–3 years of age (Figure 1 and Appendix A). The boom in acquisition of younger dogs during the pandemic is evident, with a large number of dogs aged between 0–2 years visible (Figure 1), many of which were intact at the time of the survey.

In total, 58% of dogs in this sample were purebred, with known (27.6%) and unknown crosses (14.4%) making up the rest of the sample. The most common breed (including known crosses) in this sample was the Labrador Retriever, followed by the Cocker Spaniel, Cockerpoo, Jack Russell Terrier and Border Collie. The top 25 most common breeds, representing 71% of responses, are listed in Figure 2. The median and mean average age, respectively, by purebred status was: known cross: 56.0 months (IQR 72.0) and 64.4 months (SD ± 46.1), pure breed: 66.0 months (IQR 81.0) and 72.2 months (SD ± 49.0), and unknown cross: 77.0 months (IQR 80.0) and 81.8 months (SD ± 49.2).

#### 3.1.1. Acquisition Factors

The top three most common sources for acquiring a dog included pet-selling websites (20.6%), through family and friends (18.2%) and via a UK rehoming charity website (12.1%). Selecting ‘other’ and providing a free-text response regarding acquisition was also common (12.5%). A sample of responses was qualitatively coded and grouped with common themed responses including (but not limited to):Keeping of foster dogs—often termed ‘foster fails’ by respondentsThrough work, such as veterinary practices (PTS cases) or animal sheltersReceived from someone known to them—customer, client, neighbour, colleague but not family or friendsInherited or taken on following death of owner known to the personOther organisations—assistance dogs/guide dogs, met police, ex working dog, dog warden/local poundAbandoned/found/strayThrough a breeder previously used or known to themPrivate sale or rehome—including accidental litters, pet shops, farm littersOverseas breedersBred themselvesGift/present

Dogs in this sample were commonly acquired since the pandemic with around a quarter of the sample acquired since 2020 (26%), although our request for the survey to be initially completed for the most recently acquired dog will have biased this estimate upwards if respondents had more than one dog but only completed the survey for one dog. The majority of owners (65%) acquired their dog when they were under 1 year of age, with acquisition decreasing with increasing age across other dog age groups. Less than 5% of dogs were acquired when considered ‘senior’ (around 8 years of age and older).

The majority of respondents paid either GBP501–1000 or GBP251–500 for their dog; however, just over 15% of respondents paid GBP1001–3000. Paying greater than GBP2001 only became prevalent in the years during the COVID-19 pandemic and paying GBP1001–2000 notably increased in this period too (Figure 3).

The purebreds that most commonly cost >GBP1000 included the Miniature Dachshund (54.2%), French Bulldog (54.1%), Golden Retriever (39.4%), Boxer (34.6%) and Pug (26.1%). Common crosses costing more than GBP1000 included the poodle mixes Cockerpoo (38.0%) and Labradoodle (33.2%) (Appendix A). The cost paid for a dog also varied by source from which the owner acquired their dog (Appendix A). Between 2016 and 2019, owners acquiring their dog via ‘Charity’ sources (including the website of a UK or foreign rehoming charity or a visit to a rehoming centre) most commonly paid between GBP100 and 250. During 2020–2021 this increased to GBP251–500. For those acquiring from ‘Breeder’ (breeder website, breed group website, kennel club website breeder lists or visit to a breeder), ‘Commercial’ (pet selling websites, general selling websites, social media, local press or local adverts) and ‘Personal’ (friends or family, local community or word of mouth) sources it was most common to pay between GBP501 and 1000 in 2016–2019; this notably increased during 2020–2021 with GBP1001–2000 most commonly being paid, with an increasing number of owners paying >GBP2000 when acquiring their dog from these sources.

#### 3.1.2. Veterinary Factors

The majority of dogs in this sample are currently registered with a vet (98%). Of the remaining owners who reported their dog was not registered with a vet, 41% stated they planned to do so, and another 41% stated they would if their dog became ill.

With regards to the payment of veterinary fees, 65% of owners reported they would use insurance to cover some or all of the cost of vet care, whilst 45% said they would use money from savings. Other methods of payment were less likely to be sought, such as support from family and friends (8%),and with support from charities (2%). Owners were able to select more than one response to this question; therefore, veterinary fees may be covered by more than one option by some owners. When looking at those that selected only one option of payment, the most common means of payment were still insurance (59%) and savings (32%). Relying on support from family and friends (2%) and charity (1%) were notably lower.

### 3.2. Owner Demographics (Appendix A)

A total of 354,046 complete responses are included in the summary statistics below. The majority of households had one dog only (67%), with 25% owning two and the remaining having greater than two dogs (Appendix A). The mean number of dogs owned per household was 1.38. Dog owners in this sample were most commonly female (73.5%) and aged 45–64 (48.2%).

#### Households

Dog-owning households were most commonly formed of two people (34.9%), and the majority did not have any children and/or infants within the household with single- or multiple-adult-only households most common (74.5%) and households containing adults, children and/or infants representing just over a quarter of the sample (25.5%). In total, 19.6% of households had at least one child (aged 5–17), whilst 8.3% of households had at least one infant (aged 0–4). Having both children and infants was relatively uncommon (2.1%).

### 3.3. Patterns and Trends in Ownership

Annual peaks and troughs in acquisition were evident with dogs most commonly acquired in the summer months, and less so in winter months, with the most common month of acquisition being August, followed by September and July (Figure 4).

Patterns in breed ownership were also evident, particularly related to owner age group. Whilst overall, the most popular dog breed across all age groups was the Labrador, other breeds were favoured by different age groups within the top 25 breeds (Figure 5). In younger age groups Cocker Spaniels and Cocker mixes were also commonly owned; however, less so for those aged 75 and over, where breeds such as the Jack Russell Terrier and Border Collie were more commonly owned. Current ‘designer’ breeds such as cockerpoos and ‘fashionable’ breeds such French Bulldogs, Pugs, Miniature Dachshunds and Huskies were markedly more common in younger owner age categories.

#### COVID-19 Pandemic

The COVID-19 pandemic saw a boom in dog ownership. Differences in variables such as cost, source and age of owners were observed. Out of the total of 440,759 dogs, 102,436 were acquired since 1 March 2020 (23.2%), with 58,171 (56.8%) acquired as puppies (between 0–4 months). The cost of dogs during the pandemic spiked with the increased demand, with people paying over GBP2000 only prominent since the pandemic (2020 onwards) and GBP1–2000 markedly increased in this period too (Figure 3).

Younger people (18–34) were more likely to acquire a dog during the pandemic (OR 1.62 (95% CI 1.60–1.65)) compared to older age groups, with 48% of pandemic dogs acquired by those up to 44 years. Among those people that did acquire a dog during the pandemic, younger people were more likely to acquire a dog under 4 months of age than were people in older age groups. An increase in the numbers of 18–24-year-olds acquiring a dog between 2018–2019 and 2020–2021 was observed; however, the most common age group acquiring dogs during the pandemic was 25–34 (Figure 6).

Finally, the way in which people acquired dogs also demonstrated changes during the pandemic period. Comparing dogs acquired between 2020 –2021 to those acquired between 2018 –2019, whilst the top three sources (pet websites, family and friends and rehoming websites) remained the most common throughout, more owners were sourcing their dogs from pet websites (7.6% increase) and family friends (2.0% increase), whilst there was a reduction in the proportion sourcing from a rehoming centre via both their website (2.3% decrease) and visits to the centre (3.0% decrease). However, an increase was observed in those acquiring their dogs from an overseas rehoming centre (1.7% increase) (Appendix A).

## 4. Discussion

Dogs Trust, the UK’s largest canine welfare charity, commissioned a nationwide survey to capture an up-to-date snapshot of the changing landscape of UK dogs and their owners. This ‘National Dog Survey’ was the biggest survey of dog owners in the UK to date, including a sample of 440,423 dogs and their owners. Here we describe the demographic and acquisition-related data for this sample. We also highlight notable patterns and trends in dog ownership data over time, and particularly since the COVID-19 pandemic, which saw significant changes to the population, as well as to owner behaviour when it came to the acquisition of dogs. This dataset offers an invaluable insight into the current dog population, aiding stakeholders working in the field.

Dogs are undoubtedly firm favourites in UK households and important members of families. However, contrary to previous research that highlighted children as a potential motivator for dog ownership in the UK, as well as worldwide [4,7,23,26,31], respondents were most commonly from adult-only households, with households including children and/or infants only making up a quarter of our sample (25.5%). However, where households did feature children, those with older children rather than infants were more common in this study, as in previous research [4,7,15,21,26]. This poses questions as to whether there is a shift in current ownership to households without children, particularly with an increase seen in younger age groups owning dogs, or whether families with children were underrepresented in this sample, with adult-only households simply more likely to respond to the survey. Owning a pet can create the nurturing relationship similar to that of parent and child, with some adults favouring pet ownership over having children [32]. This could be reflected in the data within this population, where people are acquiring dogs instead of having children or are owning dogs ahead of having children later. It is also possible that dog owners may also have had children at the time of acquisition, but the children have since left home. As our data are cross-sectional, it is unclear whether respondents had children at the time of acquiring their dog, which limits conclusions on these findings. Further research is required using a representative sample across all age groups to determine whether children remain motivators for dog acquisition, or whether a shift in ownership to adult-only households is happening, particularly following the COVID-19 pandemic. Previous studies have also shown that the likelihood of dog ownership also increases with household size [4,7]; however, again, this is not reflected within our dataset, with households most commonly made up of two people. This may be explained by ownership changes as a result of the pandemic, with younger people acquiring dogs, making adult couples more likely to own dogs rather than larger households. Furthermore, data from the Office of National Statistics for households in 2020 demonstrated that the average household size is 2.4 people and around 29% of family households include dependent children [33]; therefore, our data may be more representative of the wider population. Similar to previous studies, the majority of households had one dog only (67%), and the mean number of dogs per household overall was 1.38. This is very similar to existing studies, which ranged between 1.3–1.4 [4,7]. Recent research investigating the purchasing of puppies during the pandemic found that owners acquiring their dog during the pandemic were less likely compared to those pre-pandemic to have owned a dog before [34], suggesting that many recent dog owners are likely to be first-time owners.

Understanding the demographic features of the dog population is important for measuring and predicting the overall welfare state including identifying risk factors for health, behaviour and disease outcomes (which may result in euthanasia or relinquishment) and modelling longevity. Understanding the age structure, sex and neuter status of a population aids the assessment of population dynamics, including the predicted growth of a population; thus, obtaining this information regularly can aid the provision of appropriate services such as health care. In this study we found that male dogs were more common than females, which has also been demonstrated in other datasets [31,35]. It is unclear why there are greater numbers of male dogs in the owned population, but this could arise from a greater rate of entry of males into the populations, more rapid loss of females from the population, or both; further studies are needed. Literature describing shelter populations often report more males are relinquished than females [36,37] and are more likely to take longer to rehome [38,39]. This may be due to the fact that there are simply more males in the population overall, or that male dogs are more at risk of relinquishment due to factors such as behaviour. With a higher number of males in the population, this could result in a higher burden on shelters going forward, if males continue to make up a larger proportion of the general population.

The majority of dogs in this study were neutered, similar to other UK-based studies [9,25,40]. Neutering is often considered an act of responsible ownership, seen as a necessity for the control of population numbers and preventing unwanted litters [41]. Neutering is also often discussed as a means to prevent certain health conditions and modify behaviour in the UK, with uptake around 60% of dogs overall [25,40]. However, a lower neutering rate is described in other countries worldwide [19,31]. It is common within the literature that females are neutered more than males [13,19,25,42], and this was true within our sample, with females overall being 1.4 times more likely to be neutered than males. This was the case across all age categories, but in particular females over 10 years were much more likely to be neutered than males in this age group with an odds ratio of 2.48, suggesting females are much more likely to be neutered at some point during their lives, whereas it is not uncommon for males to remain entire for their whole lives. The motivations, attitudes and behaviours surrounding owners’ decisions to neuter are complex [42] and further research is needed to better understand both the benefits and risks to the individual and population of neutering to ensure evidence-backed recommendations are made, as well as into the drivers of and barriers to decision-making of owners. The age at which pet dogs should be neutered is contested, with recent scientific literature suggesting timings of neutering should be considered on an individual basis accounting for factors including behaviour, sex and breed [43]. Historically, owners were recommended to have their dogs neutered around the age of 6–9 months or after their first season for females, which is around 12–15 months [40]; however, there is limited scientific evidence to suggest a single optimum age. In this sample, early neutering does not appear particularly prevalent, with the majority of the population aged under 1 unneutered, with neutering becoming much more prevalent between the ages of 1 and 3 in both sexes, once dogs had reached sexual maturity. It is unclear whether this is representative of the current dog-owning population, or whether this is due to the impacts of the pandemic reducing access to veterinary care [44] and thus neutering services, and therefore follow-up studies are required.

Dog breeding in the UK has long been diverse, with the UK Kennel Club currently having over 200 recognised breeds [45]. Whilst initially bred for specific behavioural functions, with time dog breeds have increasingly been developed for desirable looks, shapes and sizes that make them seemingly more suited to lifestyles as companion animals [1,46]. Previous research has shown that breed is an important factor in the acquisition process [47], with owners desiring known and recognised breeds for reasons such as appearance and assumed temperament, based on breed standards or prior experience. It is therefore unsurprising that the majority of the sample (58%) in our current study, similar to other studies [25], reported that they own a purebred dog. The most common breeds included working breeds such as the Labrador Retriever, which has long been the UK’s favourite breed [48], Cocker Spaniel, Jack Russell Terrier and Border Collie. Trends in recent years have seen the increase in breeding for looks over welfare to meet demand and ‘fashion over function’ [49], with an increase in brachycephalic breeds such as French Bulldogs, Pugs and those with other exaggerated features such as Miniature Dachshunds increasing in popularity [50,51]. Furthermore, recent times have seen a rise in the popularity of specific crosses and lines originating as cross-breeds, such as the Cockerpoo and Labradoodle, originally bred for reasons such as being ‘hypoallergenic’ and assumed temperament traits that suit people’s lifestyles [52]. Research by Packer et al. described a shift in owner acquisitions during the COVID-19 pandemic, where crossbreeds were favoured over purebreds [34]. In our sample, those that were reported as a known crossbreed were on average younger than purebreds, further supporting a recent increase in the acquisition of known crosses. Packer et al. [34] also found a recent change in the desirability of dogs bred for a specific working function (e.g., gundogs), with greater favourability shown towards breeds and crossbreeds that are perceived to be good companions, with owners citing suitable sizes to fit into owners’ lifestyles and good with children as key motivators for acquisition. It is unclear whether this is a permanent shift in preference from purebred to crosses or influenced by the conditions of the pandemic; therefore, research is needed in the coming years to monitor this trend within the population. Differences in breed ownership across age groups were also found in this study. Previous research has highlighted the potential impacts of fashion trends and media sources on breed popularity and decision to acquire certain breeds [53,54,55]. This raises potential welfare concerns with people acquiring dogs ill-suited to their lifestyle or that have cost-intensive health conditions, with the previous literature suggesting younger and first-time owners may be at greater risk of reporting undesirable behaviours and relinquishing their dogs [56]. Whilst Labradors remained the preferred breed across all age groups, we found a greater preference for fashionable and designer breeds in younger owners, who may be more exposed to these breeds through a range of factors, including social media and ‘influencers’ online.

There are currently multiple key sources and routes of acquisition, which remain mostly unregulated and largely unobserved, and a lack of responsible trade to meet demand. This can result in owners unknowingly acquiring their dog from a low-welfare source [57]. In our sample, most people acquired their dog as a puppy, with the most common means of acquisition being via pet-selling websites. Acquiring a dog through online selling sites and acquaintances is a common and growing practice of dog acquisition both in the UK and elsewhere [15,20,34]. There are a number of concerns around the use of online platforms for advertising and selling pets, often around unethical practices due to the lack of regulations around sales. With online platforms increasingly used, the ease with which a pet dog can be obtained on impulse has substantially increased [58]. Furthermore, online sales aid unscrupulous breeding methods and untrustworthy sources such as puppy farming and smuggling by allowing sellers to remain anonymous. Similar to other studies and data, rehoming a dog from a charity or organization, offering a dog a ‘second chance’, was also a common source of acquisition. It should however be noted that the responses from owners of dogs acquired from rehoming organisations may be overrepresented in our sample due to the initial targeting of survey advertising towards Dogs Trust supporters. Acquiring via friends and/or family were also common sources used by UK dog owners. However, there is limited research into this and as the practice is likely to be largely informal, it is difficult to observe [59]. Certain sources may incur welfare concerns for dogs, including inadequate health checks, preventative health care such as vaccinations and failure to microchip before being acquired by their new owner. Furthermore, research from a recent longitudinal study (‘Generation Pup’) has highlighted that many owners are acquiring their dog under the recommended age of acquisition (8 weeks) and/or without seeing the mother as recommended [60]. While there is growing knowledge of the common sources of acquisition of dogs in the UK and elsewhere, much less is understood about the processes and practices some of these involve. Further research may enable greater support to people acquiring dogs to ensure welfare is maximised.

The results of this study indicate the steadily increasing prices that owners paid for their dogs, with a particularly sharp increase noticeable since 2020. The majority of owners in our sample paid GBP500–1000 for their dog, but since the COVID-19 pandemic, it has been increasingly common to pay above GBP1000—particularly for fashionable breeds such as Miniature Dachshunds and French Bulldogs.

Failure to meet individual wellbeing and health care needs for a dog can increase the likelihood of reduced quality of life and risk of negative welfare outcomes such as relinquishment, euthanasia or damage to the human–animal bond. It is therefore important to understand the veterinary-health-advice-seeking behaviour and provision of health care to aid the understanding of and guide monitoring of the health of the overall population. Knowing the proportion of the pet population accessing and receiving vet care can also inform the volume of services needed to meet population demands. Our results showed that nearly our entire sample (98%) was registered with a vet—while within the remaining 2% which were not registered, 82% of these people planned to do so or would if their dog needed care. The affordability of vet care can pose a challenge to some owners due to the varied range of services, treatments and diagnostics available, many of which may be cost-intensive. In this study, respondents reported that payment of vet fees was most commonly by insurance or by savings, with it being less common to rely on support from other sources. Insurance can influence preventative health and health-seeking behaviour by owners for their pets, with those who insure their animal typically visiting the vet more frequently and agreeing to more complex diagnostics and treatments where applicable [25,61]. Health and disease research using primary care data frequently shows insured dogs have higher odds of receiving a diagnosis for health complaints compared to non-insured dogs [62,63,64]. This raises concerns that dogs may have differential access to necessary vet care based on how their owners fund their vet care, as well as other factors, and may result in diminished welfare and even unnecessary euthanasia of dogs. More research in this area is needed to determine UK dog owners’ health-seeking behaviours and access to veterinary care based on their financial resources and other factors. 

Between 2020 and 2021, there has been a reported boom in dog ownership in the UK, making research into the current dog and owner demographics and the subsequent needs of the population even more timely. A quarter of the dogs in this study were acquired since the onset of the COVID-19 pandemic. Dogs and puppies acquired during this time may be more at risk of behaviour problems due to a lack of socialisation linked to the conditions and restrictions created by pandemic lockdowns or reduced owner time once restrictions eased and owners returned to work and workplaces [65,66]. Our data also highlight seasonal peaks and troughs in acquisitions, with summer months being the most common time of year to acquire a dog. For many, the conditions that lockdown and restrictions created, such as more time spent at home, provided a suitable opportunity to introduce a dog to their household, whether this was planned in advance or by impulse. The pandemic also saw an increase in younger people acquiring dogs, and those that did were more likely to acquire a puppy. Research has highlighted that younger and first-time owners may be more at risk of mismatched expectations with regards to ownership and therefore more likely to report problems such as undesirable behaviours [42,56,67] that can lead to relinquishment. Future research is therefore needed to determine the impact of this boom in young and first-time owners, offering appropriate interventions where possible to reduce the likelihood of negative fallout.

Of note in this study was the increase in overseas acquisition sources during 2020 and 2021. An increase in owners sourcing their dog overseas (or a puppy that has a passport suggesting it may have travelled from abroad) has been reported, often due to the demand in the UK outstripping the supply [34]. In our overall sample, 2.7% of dogs were acquired via overseas rehoming organisations; however, an increase was seen in acquiring via this route between 2018–2019 (3.4% of dogs acquired in these years) compared to 2020–2021 (5.1% of dogs acquired in these years)—an overall 1.7% increase. Overseas sellers and rescues often offer dogs that are available much more quickly than UK sources, therefore creating an attractive option to many owners wanting to acquire a dog rapidly. However, research has highlighted various associated risks with importing dogs from overseas including health and behavioural concerns [68]. People’s acquisition choices and behaviours can result in compromised welfare for both dog and owner. It is paramount that we understand these behaviours in order to inform how best to influence them, and further research is needed in this area including the development and monitoring of interventions promoting behaviour change towards more responsible acquisition and breeding. Research from Packer et al. [34] highlighted that many owners acquiring dogs during the pandemic did not acquire the breed they initially wanted due to demand or being too expensive. This could explain why owners turned to sources abroad, which may have offered a cheaper and quicker source. A drop in rehoming within UK charities was also recorded, which is again unsurprising given that many shelters were forced to close during lockdowns, and fewer dogs were relinquished due to existing owners having more time to spend with them. However, this may have been a factor in new owners being pushed towards potentially less-responsible sources. These results highlight a vast change in owner acquisition behaviour and the dog population associated with the COVID-19 pandemic. In order to determine the fallout from such events, as well as monitor ongoing trends, continued research studies are needed.

To the best of our knowledge, the National Dog Survey is the largest ever survey of UK dog owners, representing about 3.5% of the dog population and providing valuable insights into the UK dog population. However, despite aiming to be as representative as possible and utilising a large-scale awareness campaign, this study is not without limitations and may be impacted by biases due to use of convenience sampling. For example, this dataset may be subjected to response bias with those with a particular interest in dogs and animal welfare more likely to engage with advertising and therefore respond to the survey. Due to the survey being developed and shared by a canine welfare charity, an unknown level of social desirability bias may have occurred. Furthermore, this study was cross-sectional by design and therefore responses may have been impacted by recall bias, particularly for respondents who acquired their dogs a long time ago. The elements of dog ownership can be a personal and sometimes emotive subject; therefore, social desirability bias may impact people’s responses to this survey, particularly with this being distributed by a canine welfare charity, influencing respondents to give answers they believe are more aligned with good ownership behaviour. Because we used cross-sectional data, we were unable to make comparisons between certain variables of interest over time due to these not being fixed factors and therefore likely to have changed; therefore, longitudinal large-scale surveys are needed to make such comparisons. Similar to other studies, we had a much higher percentage of female respondents than males; therefore, these data may not be fully representative of the entire dog-owning population. Furthermore, whilst we received responses for all regions of the UK, we noted higher response rates in certain regions; responses were most common from dog owners in the Southeast and Northwest of England. However, when considering this proportionate to the current estimated UK dog population, the greatest number of responses were collected from the Southwest and North of Wales. It is unclear why certain groups may be more likely to respond, warranting further investigation; however, future studies should continue to seek samples as representative of the population as possible.

## 5. Conclusions

Dogs are the UK’s favourite pet, often sharing a symbiotic relationship with their owner, offering love, affection and companionship to each party. It is therefore important to regularly investigate the current pet dog population to identify and understand their needs and welfare state. This will allow for the most appropriate products, services, interventions and regulations to be developed, with the aim of reducing the likelihood of negative welfare outcomes such as health and behaviour issues, relinquishment or euthanasia. This paper aimed to summarise data representing the UK dog population and their owners utilising the largest known dataset to date through a representative national survey. We have described up-to-date demographic data for both dogs and their owners and highlighted patterns and trends related to ownership both longer-term and more recently since the COVID-19 pandemic. These data show both similarities and differences to existing studies, raising important questions and giving rise to future research areas. With significant changes to the dog population following the COVID-19 pandemic highlighted, this dataset serves as an up-to-date baseline for future study comparisons to continue to monitor trends and patterns of the dog population and dog owners going forwards.

## Figures and Tables

**Figure 1 animals-13-01072-f001:**
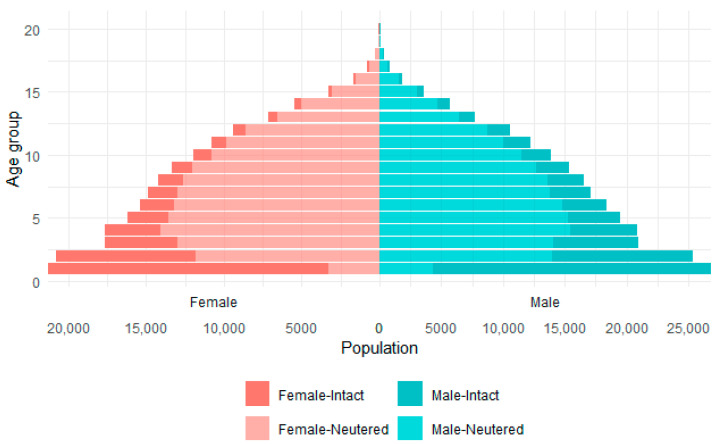
Population structure based on age, sex and neuter status of dogs in the UK.

**Figure 2 animals-13-01072-f002:**
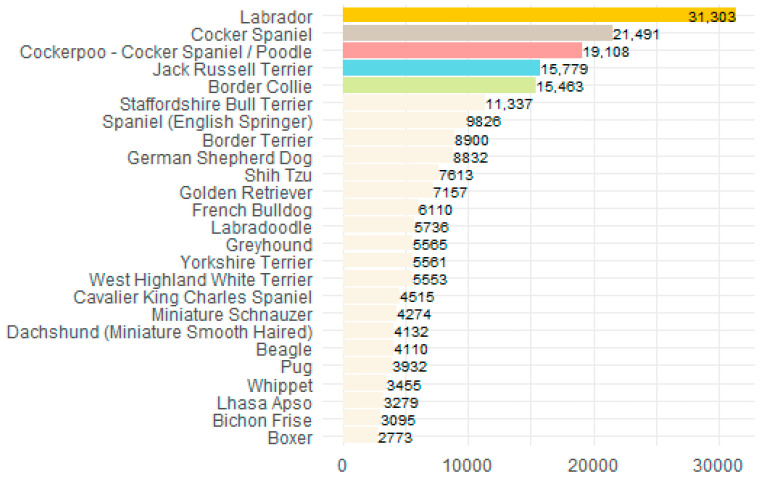
The 25 most common breeds reported by owners. The top 5 breeds are represented by individual colours for clarity.

**Figure 3 animals-13-01072-f003:**
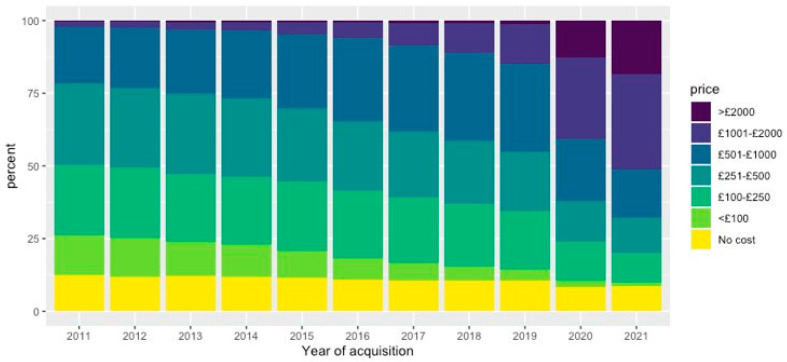
Chart summarising the price paid for dogs in the UK broken down by year.

**Figure 4 animals-13-01072-f004:**
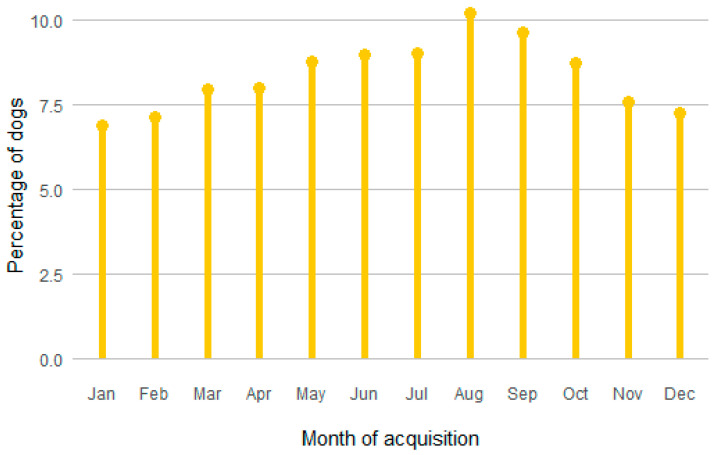
Chart showing the percentage of dogs acquired as broken down by month of acquisition.

**Figure 5 animals-13-01072-f005:**
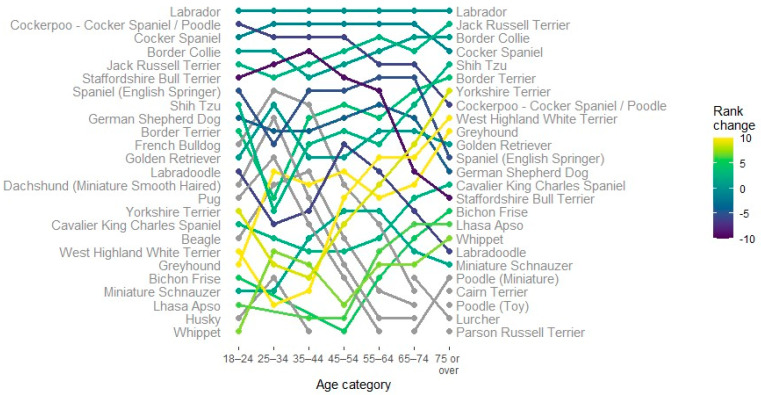
Graph showing differences in most popular breed ownership across owner age groups. Line colours represent breed rank change between the 18–24 and 75 or over age categories, with grey lines demonstrating the breed is not within the top 25 for one of these age categories.

**Figure 6 animals-13-01072-f006:**
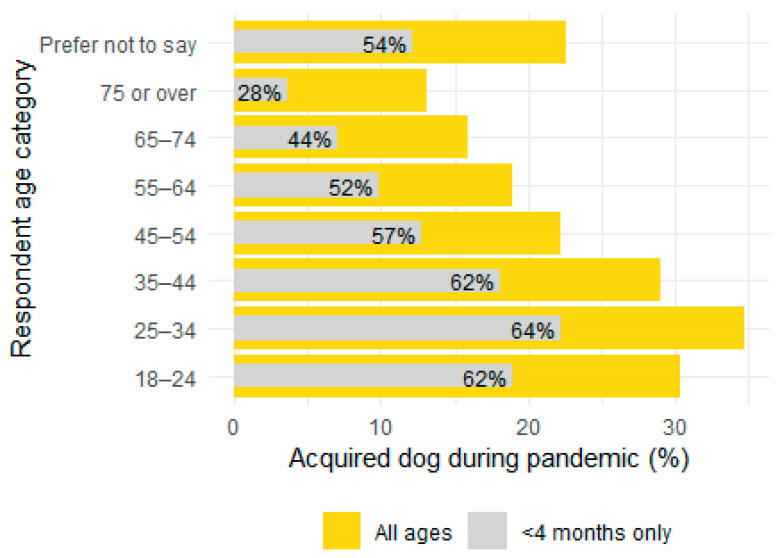
Percentage of age groups that acquired their dogs during the pandemic and of those that were acquired at less than 4 months of age .

## Data Availability

The data are not publicly available due to consent not collected from participants to share the raw data openly.

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
