# Peer review of "National Dog Survey: Describing UK Dog and Ownership Demographics"

_animals, 2023, doi:10.3390/ani13061072_

Round 1

Reviewer 1 Report

This is an excellent manuscript – the authors should be congratulated for tackling an extremely large dataset and distilling is into an acceptable and digestible set of interesting findings. My comments below are only meant to enhance an already very well written manuscript – none of these are essential, all are only suggestions.

Line 39 – delete “being popular figures in households and” – not necessary and I am not entirely sure what is meant by a ‘popular figure’?

Line 104 and 106 – not sure Creative Agency nor Digital Experience Agency needs to capitalised? I would capitalise the name of each of these agencies, but not their ‘type’.

Section 2.1 – can you clarify whether Dog Trust’s logo was visible in the survey? i.e. would the respondent have been aware that a canine welfare charity was leading this survey? Or was an attempt made to keep the survey lead anonymous?

Line 132 – I don’t know was ‘social targeting of adverts’ is – can you provide a very brief explanation? Perhaps juts in brackets after this term for luddites like me?!

Line 179-183 – Its unclear from this sentence whether females are more likely to be neutered than males in all age groups. Younger and older age groups are highlighted, is that because in the age groups in-between there is no sex bias in neutering?

Line 187 and 396 – You particularly highlight the age group 2-3 years, but actually there is also a big jump in the likelihood of neutering in the age 1-2 years as compared to 0-1 years. It looks from figure 1 that the biggest increase in neutering is when you move into the 1-2 years ago bracket. So I think I would talk about the 1-3 age range as when most neutering occurs, rather than leave out the 1-2 year group.

Line 192 – It would be ideal to include the % of known and unknown crosses here, not just the % of purebreds – in particular, I would have like to see how many unknown crosses there are in the UK population.

Section 3.1.1 – I appreciate that the data is available in table S1 – but I would be tempted to bring the % of acquisition from overseas charity/rehoming sources into the main text, only because it is talked about in some detail in the discussion. This may mean that you need to include a graph of the top 10+ acquisition routes (this would also allow people to see that you separated pet selling websites from kennel club breeder websites and individual breed websites, which they may not immediately appreciate). However, this would lengthen the text a lot, so please only consider this a suggestion. Perhaps a shorter alternative would be to include the datapoint 2.7% acquired from overseas charities/rehoming when you talk about this in the discussion? Related to this is how you report the change in overseas rehoming, in line 524 you describe this source as increasing in 2002/2021 – perhaps include the relevant data from S9 here? so 3.4% 18/19 to 5.1% 20/21, an increase of 1.7% (from S9). I know that you state this 1.7% increase in line 310, but it’s a 1.7% from what baseline in 18/19? This would help the reader understand this key point about the size and change in this acquisition route more easily, without having to search the supplementary tables.

Section 3.2.1 – Perhaps include the % of household that were families? Is that the combination of older children 19.6%, infants 8.3% and both 2.1%? But that seems to many if 74.5% are adult only?

Line 281 – Do you mean other breeds were favoured “by different” age groups? Rather then “across” age groups?

Figure 5 – This is a great graph – I like it a lot! However there are some points towards the bottom on the graph where the breed is not possible to distinguish – In particular for age groups in the middle – because they do not feature in the top 25 of either the youngest or oldest owners. Might be easier to just delete these points? Although, they are not too distracting, so again, please consider this a suggestion to ignore if its not easily achievable.

Line 299 – “among those people that did acquire a dog during the pandemic, younger people were more likely to acquire a dog under 4 months of age” – is that more likely than owners in other age groups? i.e. finish with “…than people in older age groups”

Figure 6 – Can you please clarify – I am not sure if 30% of 18-24 year old owners acquired a dog during the pandemic, or of those people that acquired a dog during the pandemic, 30% were 18-24?

Line 316 – Suggest this is “of” UK dogs not “for” UK dogs

Line 331 – Can you compare your % of households that are a family to the ONS data referenced in line 348? This would help understand if you sample is representative?

Line 382 – Suggest a rephase to emphasise that it is the greater odds ratio (2.48) in the older females as compared to females in general (1.4) that is the interesting point, not the statistical significance.

Line 421 – Is Packer’s point that desirability of working dog breeds has decreased not just changed, so although they are still currently the most popular, there is increase in breeds/crossbreeds perceived to be good companions etc

Line 448 – suggest deleting “tapping into consumer demand for fashionable breeds that come with a high price tag whilst” and replace with “by”

Line 466 – I would be tempted to make a clearer comment on the startling jump in prices – there is a steady increase in prices throughout the years and then an astonishing jump in 2020 and 2021.

Line 469-480 – Consider combining this with the points made in the paragraph starting 524.

Line 510 – where does the 3% increase in acquisitions come from? I don’t see it in the results section – is this from looking the age bins in figure 1 or year of acquisition in S1? Perhaps could explain more in line 187-189?

Author Response

This is an excellent manuscript – the authors should be congratulated for tackling an extremely large dataset and distilling is into an acceptable and digestible set of interesting findings. My comments below are only meant to enhance an already very well written manuscript – none of these are essential, all are only suggestions.

Thank you for taking the time to review this manuscript and for your positive and constructive feedback!

Line 39 – delete “being popular figures in households and” – not necessary and I am not entirely sure what is meant by a ‘popular figure’?

This has been removed!

Line 104 and 106 – not sure Creative Agency nor Digital Experience Agency needs to capitalised? I would capitalise the name of each of these agencies, but not their ‘type’.

Agree – this has been amended, thank you!

Section 2.1 – can you clarify whether Dog Trust’s logo was visible in the survey? i.e. would the respondent have been aware that a canine welfare charity was leading this survey? Or was an attempt made to keep the survey lead anonymous?

This has been amended to make this clearer – the survey featured regular Dogs Trust branding and was always advertised as being conducted by Dogs Trust therefore respondents are highly likely to be aware this was lead by a canine welfare charity. We have added a comment to the limitations to highlight this could also lead to some bias in/influence peoples responses. (See line number 578-579).

Line 132 – I don’t know was ‘social targeting of adverts’ is – can you provide a very brief explanation? Perhaps juts in brackets after this term for luddites like me?!

We have added a comment in brackets explaining this is towards targeted demographic groups

Line 179-183 – Its unclear from this sentence whether females are more likely to be neutered than males in all age groups. Younger and older age groups are highlighted, is that because in the age groups in-between there is no sex bias in neutering?

We agree this was unclear, and have added further clarification to this section to demonstrate that neutering is more common in females across all age groups.

Line 187 and 396 – You particularly highlight the age group 2-3 years, but actually there is also a big jump in the likelihood of neutering in the age 1-2 years as compared to 0-1 years. It looks from figure 1 that the biggest increase in neutering is when you move into the 1-2 years ago bracket. So I think I would talk about the 1-3 age range as when most neutering occurs, rather than leave out the 1-2 year group.

Agreed – thank you for highlighting this. We’ve amended both those sections to 1-3 years now.

Line 192 – It would be ideal to include the % of known and unknown crosses here, not just the % of purebreds – in particular, I would have like to see how many unknown crosses there are in the UK population.

These have been added in!

Section 3.1.1 – I appreciate that the data is available in table S1 – but I would be tempted to bring the % of acquisition from overseas charity/rehoming sources into the main text, only because it is talked about in some detail in the discussion. This may mean that you need to include a graph of the top 10+ acquisition routes (this would also allow people to see that you separated pet selling websites from kennel club breeder websites and individual breed websites, which they may not immediately appreciate). However, this would lengthen the text a lot, so please only consider this a suggestion. Perhaps a shorter alternative would be to include the datapoint 2.7% acquired from overseas charities/rehoming when you talk about this in the discussion? Related to this is how you report the change in overseas rehoming, in line 524 you describe this source as increasing in 2002/2021 – perhaps include the relevant data from S9 here? so 3.4% 18/19 to 5.1% 20/21, an increase of 1.7% (from S9). I know that you state this 1.7% increase in line 310, but it’s a 1.7% from what baseline in 18/19? This would help the reader understand this key point about the size and change in this acquisition route more easily, without having to search the supplementary tables.

Thank you for this. We have opted to include this in the discussion highlighting the overall 2.7% of the sample acquiring via this route, then breaking this down to compare the 18/19 and 20/21 groups! (Now on lines 546-551)

Section 3.2.1 – Perhaps include the % of household that were families? Is that the combination of older children 19.6%, infants 8.3% and both 2.1%? But that seems to many if 74.5% are adult only?

This section was a little bit unclear so we have included further percentages to hopefully highlight this better – thanks for flagging!

Line 281 – Do you mean other breeds were favoured “by different” age groups? Rather then “across” age groups?

Yes we did – amended!

Figure 5 – This is a great graph – I like it a lot! However there are some points towards the bottom on the graph where the breed is not possible to distinguish – In particular for age groups in the middle – because they do not feature in the top 25 of either the youngest or oldest owners. Might be easier to just delete these points? Although, they are not too distracting, so again, please consider this a suggestion to ignore if its not easily achievable.

The figure has been updated to hopefully increase the clarity and improve the graph!

Line 299 – “among those people that did acquire a dog during the pandemic, younger people were more likely to acquire a dog under 4 months of age” – is that more likely than owners in other age groups? i.e. finish with “…than people in older age groups”

Thank you – this has been added.

Figure 6 – Can you please clarify – I am not sure if 30% of 18-24 year old owners acquired a dog during the pandemic, or of those people that acquired a dog during the pandemic, 30% were 18-24?

Apologies – this is 30% of owners aged 18-24 years acquired their dog during the pandemic. Figure 6 legend has been updated to hopefully make this clearer

Line 316 – Suggest this is “of” UK dogs not “for” UK dogs

Thank you – amended!

Line 331 – Can you compare your % of households that are a family to the ONS data referenced in line 348? This would help understand if you sample is representative?

We’ve added some sentences around this, (line 364) demonstrating our sample in comparison to the 2020 ONS data.

Line 382 – Suggest a rephase to emphasise that it is the greater odds ratio (2.48) in the older females as compared to females in general (1.4) that is the interesting point, not the statistical significance.

This section has been tweaked to reflect this

Line 421 – Is Packer’s point that desirability of working dog breeds has decreased not just changed, so although they are still currently the most popular, there is increase in breeds/crossbreeds perceived to be good companions etc

Packers findings highlight that owners acquiring their dog during the pandemic cited acquiring a dog for specific function (such as a working dog) as a motivator for acquisition less frequently than other factors (such as good companions and suitable size for lifestyle). We have added in some further explanation to make this clearer in the text.

Line 448 – suggest deleting “tapping into consumer demand for fashionable breeds that come with a high price tag whilst” and replace with “by”

Amended

Line 466 – I would be tempted to make a clearer comment on the startling jump in prices – there is a steady increase in prices throughout the years and then an astonishing jump in 2020 and 2021.

This sentence has been amended to highlight a steady increase in prices over time with a particularly sharp increase since 2020.

Line 469-480 – Consider combining this with the points made in the paragraph starting 524.

We agree this is a more appropriate place to include this, so have moved to this section – thank you.

Line 510 – where does the 3% increase in acquisitions come from? I don’t see it in the results section – is this from looking the age bins in figure 1 or year of acquisition in S1? Perhaps could explain more in line 187-189?

This was from an earlier draft of the paper, which we have now removed – the 3% increase was the difference in those acquired between 2019 and 2020, however as we asked people to respond with their most recently acquired dog, we felt this wasn’t an appropriate statement to demonstrate an increase in ownership during the pandemic (as it may be interpreted as), and so have discarded this from the current version of the manuscript – thanks for flagging!

Reviewer 2 Report

It was a real pleasure to be invited to review this manuscript on UK dog ownership and their demographics. It is very well written, contains a huge sample set, and some very important and interesting data and I will be delighted to see it published.

I have a few very minor comments, and a few questions, which the authors can feel free to ignore if they wish

The data from lines 207-220- I wonder if it would be worth adding some numbers in here? Just to give the reader an idea as to how common they were. Particularly with the importations which are reported to have increased during and post covid with the health risks to humans an animals that may bring.

Also, do UK pet shops sell dogs? I thought that this practise was banned? (Excuse my ignorance of UK law)

Figure 4- does this figure alter with year post covid? Again, feel free to ignore

Line 324- I think that you can delete ‘as’ here

Line 489- would ‘2% which was not registered….’ Perhaps read a bit better?

Line 542- ‘…respond to the survey ….’ Maybe?

Line 552- it may be worth talking through this a bit more- where did you get higher responses from? Cities? Were certain areas more heavily affected by covid?

But once again, my congratulations to the authors, and please excuse my  ignorance in parts. I wish you every success for the future, and thank you for making a reviewers life easy!

Author Response

It was a real pleasure to be invited to review this manuscript on UK dog ownership and their demographics. It is very well written, contains a huge sample set, and some very important and interesting data and I will be delighted to see it published.

Thank you very much for taking the time to review and for your positive comments!

I have a few very minor comments, and a few questions, which the authors can feel free to ignore if they wish

The data from lines 207-220- I wonder if it would be worth adding some numbers in here? Just to give the reader an idea as to how common they were. Particularly with the importations which are reported to have increased during and post covid with the health risks to humans an animals that may bring.

Unfortunately given this size of the dataset, in order to fully quantify these free text responses accurately and thus provide quantification of how common they are in the overall dataset, it would be incredibly time consuming which we felt wasn’t justified by what it would provide to the paper. A subset of these were analysed and coded qualitatively to provide some of the more prevalent ‘themes’ of answers and therefore we feel quantification based on this type of analysis would not be accurate and could be misleading so omitted to include numbers, and just highlighted the theme areas.

Also, do UK pet shops sell dogs? I thought that this practise was banned? (Excuse my ignorance of UK law)

It became illegal within the UK to sell pets through shops following legalisation brought in in 2018, and therefore the practice before this was able to be carried out. As we have many dogs in our sample acquired before 2018 it is possible for them to have been acquired via this route legally– hope that clarifies things!

Figure 4- does this figure alter with year post covid? Again, feel free to ignore

We have not analysed this specifically year by year, and chose to include this averaged across as all years we have data collected for.

Line 324- I think that you can delete ‘as’ here

Deleted – thank you!

Line 489- would ‘2% which was not registered….’ Perhaps read a bit better?

Thank you – amended!

Line 542- ‘…respond to the survey ….’ Maybe?

Amended

Line 552- it may be worth talking through this a bit more- where did you get higher responses from? Cities? Were certain areas more heavily affected by covid?

We have included some further points here incorporating the regional data we collected to highlight where differences occurred.

But once again, my congratulations to the authors, and please excuse my  ignorance in parts. I wish you every success for the future, and thank you for making a reviewers life easy!

Your positive feedback is greatly appreciated!

Reviewer 3 Report

Authors aims to identify characteristics of dog-owning households, describing the dog and owner demographics of the UK dog population and to describe changes in demographics of ownership and dogs over time with also a view on the COVID-19 pandemic.

The manuscript is of very high quality, perhaps the only thing to question is why not to search of other research question and therefore other more deep analysis perhaps a multivariable approach, considering the volume of data available.

Author Response

Authors aims to identify characteristics of dog-owning households, describing the dog and owner demographics of the UK dog population and to describe changes in demographics of ownership and dogs over time with also a view on the COVID-19 pandemic.

The manuscript is of very high quality, perhaps the only thing to question is why not to search of other research question and therefore other more deep analysis perhaps a multivariable approach, considering the volume of data available.

As this paper is intended as a descriptive overview of the survey itself and the main demographic results from the survey, we felt it inappropriate to delve into a few more specific research questions only and potentially miss some important findings. We feel as it stands, this paper acts as a hypothesis and future research question generator to guide both future research from this data set – including multivariable models as suggested- as well as to other researchers. As this dataset is so large, applying such statistical approaches to the full data is likely to return statistical significance where the operational significance may be limited, and therefore more targeted analysis of variables of interest would be warranted. We also feel adding further results like this to this paper would cause a significant increase in it’s length making it substantially larger and not as digestible as we had intended it to be.